# Pseudorabies Virus Mutants Lacking US9 Are Defective in Cytoplasmic Assembly and Sorting of Virus Particles into Axons and Not in Axonal Transport

**DOI:** 10.3390/v15010153

**Published:** 2023-01-04

**Authors:** Steven Adamou, Adam Vanarsdall, David C. Johnson

**Affiliations:** 1Multiscale Microscopy Core, Oregon Health & Science University, Portland, OR 97239, USA; 2Department of Molecular Microbiology & Immunology, Oregon Health & Science University, Portland, OR 97239, USA

**Keywords:** neurons, anterograde transport, axons, gE/gI, assembly, sorting, kinesins

## Abstract

Herpes simplex virus (HSV) and varicella zoster virus (VZV) rely on transport of virus particles in neuronal axons to spread from sites of viral latency in sensory ganglia to peripheral tissues then on to other hosts. This process of anterograde axonal transport involves kinesin motors that move virus particles rapidly along microtubules. α-herpesvirus anterograde transport has been extensively studied by characterizing the porcine pseudorabies virus (PRV) and HSV, with most studies focused on two membrane proteins: gE/gI and US9. It was reported that PRV and HSV US9 proteins bind to kinesin motors, promoting tethering of virus particles on the motors, and furthering anterograde transport within axons. Alternatively, other models have argued that HSV and PRV US9 and gE/gI function in the cytoplasm and not in neuronal axons. Specifically, HSV gE/gI and US9 mutants are defective in the assembly of virus particles in the cytoplasm of neurons and the subsequent sorting of virus particles to cell surfaces and into axons. However, PRV US9 and gE/gI mutants have not been characterized for these cytoplasmic defects. We examined neurons infected with PRV mutants, one lacking both gE/gI and US9 and the other lacking just US9, by electron microscopy. Both PRV mutants exhibited similar defects in virus assembly and cytoplasmic sorting of virus particles to cell surfaces. As well, the mutants exhibited reduced quantities of infectious virus in neurons and in cell culture supernatants. We concluded that PRV US9 primarily functions in neurons to promote cytoplasmic steps in anterograde transport.

## 1. Introduction

α-herpesviruses such as herpes simplex virus (HSV) and varicella zoster virus (VZV) infect mucosal epithelium or skin and spread into neuronal axons. The viruses then travel in the retrograde direction within axons to neuron cell bodies present in ganglia where life-long latency is established. These viruses periodically reactivate from latency and are transported in the anterograde direction in neuronal axons to mucosal surfaces or the skin where the virus replicates and produces epithelial lesions. Anterograde transport involves transport of virus particles by kinesin motors that move virus particles by fast axonal transport on microtubules (reviewed in [1,2]). Anterograde transport is an indispensable process for α-herpesviruses permitting periodic egress from neurons, spread within epithelial tissues and production of infectious virus that can disseminate to other hosts.

Many of the recent studies investigating HSV and PRV anterograde transport have focused on two membrane proteins gE/gI and US9 that promote this transport. US9 is a fascinating protein because it has largely or entirely neuron-specific effects for both PRV and HSV, i.e., US9-null mutants have defects in anterograde spread of viruses, but not in retrograde spread or in epithelial tissues or non-neuronal cultured cells [3,4,5,6,7,8]. In contrast, gE/gI (a heterodimer of two glycoproteins gE and gI) affects anterograde transport but gE/gI also promotes cell-to-cell spread in epithelial tissues [9,10,11]. The axons of neurons infected with HSV and PRV mutants lacking gE/gI or US9 contain fewer virus particles [5,8,12,13,14]. With PRV, loss of US9 alone reduced the numbers of virus particles in axons by ~80–90% [12,15]. However, with HSV, gE- or US9-mutants each showed ~50% reduced anterograde transport, whereas a gE-/US9-double mutant exhibited 90–95% reductions in virus particles present in axons [5,8].

Evidence was presented that PRV US9 is required for anterograde transport because US9 bound to kinesin motors, specifically KIF1A [16,17]. PRV gE/gI was also involved in this binding and transport [18]. Silencing of KIF1A reduced PRV anterograde spread substantially [16]. It was also reported that drug-induced artificial recruitment of KIF1A was sufficient to rescue anterograde transport of a PRV US9 mutant and US9 altered the velocity of anterograde transport in axons [17]. Thus, models were proposed that PRV US9 and gE/gI tether enveloped PRV particles onto kinesin motors in axons to promote anterograde transport. There was also evidence that HSV US9 could bind to another class of kinesin proteins, the KIF5 proteins, and it was also suggested that this binding promotes anterograde transport within axons [19,20].

However, there have been reports that HSV and PRV US9 and gE/gI are not required for kinesin-mediated transport in axons. First, a HSV double mutant lacking both gE and US9 transported very few viral particles into axons, but the few particles that entered axons were transported normally; there was no reduction in the rates of transport or stuttering or lags in transport [8]. Second, a PRV mutant lacking US9 exhibited major reductions in virus particles found in axons, but again there were normal velocities of axonal transport [15]. These observations with both HSV and PRV strongly argue that the US9 and gE/gI proteins are not important for the tethering of virus particles onto kinesin motors within axons. Instead, electron microscopic (EM) studies of neurons infected with the HSV gE-/US9- and gE-null mutants provided another model for how gE/gI and US9 function in anterograde transport [21]. HSV gE-/US9 mutant exhibited major defects in virus assembly and sorting. There was extensive accumulation of unenveloped capsids, partially enveloped capsids and fully enveloped virions in the cytoplasm of several types of neurons. Moreover, an HSV lacking just gE produced almost normal quantities of enveloped capsids, but these accumulated in the cytoplasm of neurons and did not reach cell surfaces. Extending these studies, we showed that gE/gI and US9 bind numerous tegument proteins and loss of gE/gI and US9 causes tegument protein UL16 to be degraded in cells [22]. Tegument proteins normally play important roles in secondary envelopment and the sorting of virus particles in the cytoplasm. Therefore, these studies argued that HSV gE/gI and US9 function primarily in the neuron cell bodies and not in axons. 

There are apparently at least three steps in the anterograde transport of HSV and PRV: (i) assembly of enveloped virus particles in the cytoplasm, (ii) sorting of these virus particles into axons and (iii) once in axons, the tethering of viruses onto kinesin motors for transport. For HSV, we now know that gE/gI and US9 promote assembly, cytoplasmic sorting of virus particles to cell surfaces and into axons, but these proteins are not needed for kinesin-mediated transport within axons. However, for PRV the roles of US9 and gE/gI remain unclear. On one hand, there are reports that US9 can influence the velocity of virus particles moving in axons [17]. On the other hand, a PRV US9 mutant exhibited normal anterograde transport in axons [15]. Based on these different observations, we felt that it was important to perform EM studies on neurons infected with two PRV mutants: one lacking US9 and gE/gI and the other lacking only US9. These studies produced evidence of major defects in virus assembly and cytoplasmic sorting of virus particles, defects that can fully explain the reduced numbers of PRV particles in axons and arguing against models in which US9 functions within axons [16,17].

## 2. Materials and Methods

**Viruses.** PRV strains Becker, BaBe and 161 [3] were kindly provided by Lynn Enquist (Princeton University). PRV Becker is a wild-type virus. PRV BaBe contains the deletion found in PRV Bartha that includes the genes for US2, gI (US7) gE and US9 and was constructed by transferring the deletion into PRV Becker [4,18,23,24]. The US2 gene was shown to have no effects on PRV anterograde transport [18]. PRV 161 was constructed by producing a 258 bp deletion affecting amino acids 4 to 89, which is most of the US9 open reading frame [24]. PRV 161 expressed gE/gI and other glycoproteins and analysis of the US component of PRV 161 indicated no mutations surrounding the US9 gene. These PRV strains were expanded and titered using PK15 swine kidney epithelial cells grown in Dulbecco’s modified Eagle medium (DMEM) supplemented with 10% of fetal bovine serum (FBS) with penicillin and streptomycin.

**Neuronal cell cultures.** CAD cells [25] are a derivative of a mouse catecholaminergic central nervous system cell line and were a kind gift from Greg Smith at Northwestern University Medical School, Chicago, IL and were maintained in DMEM/F12 containing 8% fetal bovine serum (FBS) and passed by gentle dissociation with sodium citrate buffer (134 mm KCl, 15 mM Na citrate, pH 7.3 to 7.4). Differentiation of CAD cells into neurons was achieved by plating cells on poly-D-lysine (30 μg/mL) and laminin (2 μg/mL) coated glass cover slips in differentiation media (DMEM/F12 containing 0.5% FBS, 10 uM 3-isobutyl-1-methylxanthine (IBMX), 150 uM dibutyryl-cAMP (dbcAMP), and 1 ng/mL nerve growth factor (NGF, 2.5S, Invitrogen). After 2 days, the IBMX and dbcAMP were removed. The cells were differentiated for 7–10 days before being infected with PRV for 10–12 h using 5 PFU/cell. Rat hippocampal neurons were purchased from Lonza Bioscience or Thermofisher. Hippocampal cells were plated on Thermanox coverslips plates at ~80,000 cells/coverslip as described [21] in PNGM^TM^ BulletKit^TM^ (CC-4461, Lonza) media or Neurobasal Plus media with GlutaMax and B27 supplements (Thermofisher) as described by the providers. After 10–12 days in culture, the neurons were infected with PRV using 5 PFU/cell 10–12 h before being fixed and processed for EM. 

**Quantification of infectious virus produced by hippocampal neurons infected by PRV mutant viruses.** Hippocampal neurons in 24-well dishes were infected with PRV Becker, BaBe or 161 by using 5 PFU/cell for 2 h then the virus was removed and the cells washed twice with growth media. The cells were incubated for a total of 12 or 24 h in media before the media were removed and stored at −70 °C. The cells were scraped into media and sonicated and stored at −70 °C. The cell-associated virus and virus in media were diluted in DMEM containing 2% FBS and infectious virus determined using plaque assays involving PK15 cells incubated in DMEM containing 2% FBS and 1% methocel. After 48 h, cells were fixed with 4% paraformaldehyde, stained with 1% crystal violet and plaques counted. 

**Electron microscopy.** PRV-infected CAD neurons were washed once in 0.1 M sodium cacodylate buffer, pH 7.2, and fixed in Karnovsky’s solution (cacodylate buffer containing 2% [wt/vol] paraformaldehyde and 2.5% [wt/vol] glutaraldehyde) for 30 min at room temperature, removed from the tissue culture dish using a cell scraper, centrifuged into Eppendorf tubes, and stored at 4 °C before processing for EM [21]. Immediately prior to processing for EM, the CAD neuron cell pellet was enrobed in 75 uL of 2.5% aqueous agarose [wt/vol] heated to 75 °C. After allowing the samples to cool on ice for 30 min, a microwave protocol was used to prepare the sample for TEM as follows: Samples were rinsed in 0.1 M sodium cacodylate buffer, incubated in reduced osmium tetroxide (1.5% potassium ferrocyanide in 2% OsO4), rinsed in water, and stained *en bloc* with aqueous 1% uranyl acetate; following the uranyl acetate incubation, the CAD neurons were dehydrated in an aqueous series of 50%, 75%, 95%, and 100% acetone; Epon-812 resin infiltration was facilitated by incubation in a 1:1 solution of 100% acetone and freshly made Epon-812 resin which was followed by 4 exchanges of 100% freshly made Epon-812 resin. Samples were then transferred into coffin molds, filled with freshly made Epon-812 resin, and cured at 60 °C for 36 h. Rat hippocampal neurons attached to lysine/laminin-coated 13-mm Nunc™ Thermanox™ coverslips were infected with PRV and then the cells were washed and fixed with Karnovsky’s solution as above. Following fixation, a microwave protocol was used to prepare the sample for TEM as follows: Samples were rinsed in 0.1 M sodium cacodylate buffer, incubated in reduced osmium tetroxide (1.5% potassium ferrocyanide in 2% OsO4), rinsed in water, and stained en bloc with aqueous 1% uranyl acetate; following the uranyl acetate incubation, the hippocampal neurons were dehydrated in an aqueous series of 50%, 75%, 95%. and 100% ethanol; LX-112 resin infiltration was facilitated by incubation in a 1:1 solution of 100% ethanol and freshly made LX-112 resin, followed by 4 exchanges of 100% freshly made LX-112 resin. Samples were then transferred into embedding capsules (Beem), filled with freshly made LX-112 resin, and cured at 60 °C for 36 h. Thin sections (70 nm) were obtained from the block face of both sample sets outlined above and placed on 200 mesh Ted Pella bare copper grids (Ted Pella No: 4406). Grids were then post-stained with 5% uranyl-acetate and Reynold’s lead citrate. Grids were rinsed with water immediately following the uranyl-acetate staining and with 0.01 M sodium hydroxide and water after staining with Reynold’s lead citrate. Micrographs were captured at 120 kV on an FEI Tecnai™ Spirit TEM system using the AMT software interface on a NanoSprint12S-B cMOS camera system.

## 3. Results

**A PRV mutant lacking gE/gI and US9 failed to assemble and sort virus particles in mouse CAD neurons.** The PRV Bartha strain contains a deletion in the US component that removes the US2, US7 (gI), US8 (gE) and US9 genes. A fragment of the Bartha US component DNA was recombined into wild-type PRV Becker to produce the same deletion in a PRV named BaBe that lacks the US2, US7 (gI), US8 (gE) and US9 genes US7 (gI) [3,4,18,23]. The adjacent genes gD and gG were not affected by this recombination and US2 gene product was shown to have no effect on anterograde transport [18]. BaBe displayed neuron-specific defects specifically in anterograde spread of PRV, and not in replication in the eye or retrograde spread. Thus, BaBe is a PRV that lacks gE/gI and US9 and, in that sense, is similar to our HSV gE-/US9-double mutant, which we found to have defects in assembly and sorting of virus particles [8,21]. Mouse catecholaminergic central nervous cells (CAD neurons) can be differentiated to produce axons or neurites after 7–10 days [25]. CAD neurons were used previously in our studies of the HSV gE-/US9-mutant [21]. These neurons were infected with PRV BaBe or wild-type Becker for 10–12 h, then the cells removed from dishes, pelleted, fixed and processed for EM, as described [21]. 

CAD neurons infected with Becker displayed copious cell surface particles (Figure 1A–C), but there was also evidence of less numerous cytoplasmic enveloped particles and unenveloped capsids (see Figure 1C). Higher magnification images of Becker-infected CAD neurons showed enveloped virions present on cell surfaces, as well as enveloped virions in cytoplasmic vesicles (Figure 2A,B). CAD neurons infected with BaBe displayed fewer cell surface virions and much more numerous cytoplasmic enveloped and unenveloped particles (Figure 3A–C). Some neurons showed no obvious cell surface virions (Figure 3A) while others had some cell surface virions, although in some cases these particles exhibited defects in assembly (Figure 3B). Higher magnification images of BaBe-infected CAD neurons showed accumulation of fully enveloped virions that appeared normal, as well as malformed or misassembled virions of two types: (i) membrane vesicles surrounding several PRV capsids (Figure 4B) and (ii) cytoplasmic vesicles containing enveloped virions with electron-dense material present at one pole of the virion (Figure 4A,B, stars). There was also a significant accumulation of partially enveloped capsids (Figure 3B,C asterisks) and numerous unenveloped capsids (Figure 4A,B, filled arrows). We had previously observed substantially increased numbers of partially enveloped and unenveloped cytosolic particles with the HSV gE-/US9-double mutant [21], consistent with defects in secondary envelopment.

We counted virus particles enumerating (i) cell surface enveloped particles, (ii) cytoplasmic enveloped particles, (iii) capsids that were partially enveloped and iv) unenveloped capsids in the cytoplasm. We did not include standard deviations for each of the percentages of these virus particles. As noted before [21,26,27], there were markedly different numbers of virus particles comparing one cell to another. Thus, it makes little sense to compare how cells differ one to another. Instead, we report the percentage of particles of each type from counts of ~1000 particles involving >10 representative cells. CAD neurons infected with wild-type Becker displayed primarily cell surface virions, there was 3.3-fold more enveloped particles found on cell surfaces compared with the total number of particles in the cytoplasm (the sum of enveloped, partially enveloped, and unenveloped particles) (Table 1). By contrast, BaBe infected CAD neurons showed 7.9-fold more cytoplasmic particles, compared with cell surface enveloped virions. Moreover, there was a 7-fold decrease in the quantities of enveloped virions on the surfaces of BaBe-infected CAD neurons, compared with Becker-infected neurons (Table 1). In addition, there were 3.6-fold more partially enveloped and unenveloped capsids in the cytoplasm of BaBe-infected neurons, compared with that in Becker-infected cells. Together, these data suggested that mutations in BaBe reduced assembly of enveloped virions in the cytoplasm and caused major reductions in the delivery of enveloped particles to cell surfaces.

**Rat hippocampal neurons infected with PRV BaBe accumulated enveloped virions, partially enveloped capsids and unenveloped capsids in the cytoplasm.** We were suspicious that the reductions in enveloped virions present on the surfaces of BaBe-infected CAD neurons might underrepresent the defects caused by loss of gE/gI and US9. CAD cells were differentiated to become neurons that produce axons, but a fraction (~10%) of these cells appear less differentiated, i.e., they lacked detectable axons when characterized by light microscopy. These undifferentiated cells could be avoided in light microscopic studies. However, it was more difficult to avoid these cells in EM studies. In fact, we observed a fraction (<10%) of BaBe-infected CAD cells that displayed relatively larger numbers of cell surface enveloped virions. It is well established that HSV and PRV gE/gI and, especially, US9 have less extensive effects in non-neuronal cells. Thus, a few undifferentiated CAD cells might have skewed our results to some extent. To address this, we attempted to perform EM experiments using several other primary neurons, which are more differentiated, including rat and chicken dorsal root ganglia (DRG) neurons [28] and rat superior cervical ganglia (SCG) neurons [4,29]. In our hands, these primary neurons, produced too few PRV enveloped and unenveloped virus particles to allow counting of 100s of virus particles. By contrast, we found that rat hippocampal neurons produced higher quantities of PRV particles in EM experiments and these highly differentiated primary neuron cultures are of high purity [30,31].

Hippocampal neurons derived from embryonic rats (day 18 embryos) [32,33] were plated on poly-lysine and laminin-coated glass coverslips and incubated in Neurobasal media containing B27 supplement. After 8–12 days in culture, the hippocampal neurons were infected with PRV Becker. Note that there were not any virus particles observed in neurons infected for 2 h, supporting that the particles observed were virus progeny, not input virus. As with CAD neurons, hippocampal neurons infected with Becker displayed numerous cell surface enveloped virions (Figure 5A–C). There were also some enveloped particles and partially enveloped particles in the cytoplasm, but the majority of post-nuclear particles were on cell surfaces. Higher magnification micrographs showed enveloped cell surface virions with normal morphologies (Figure 6A,B). 

BaBe-infected hippocampal neurons displayed many fewer enveloped virions on cell surfaces (Figure 7A–C). Instead, there were many more enveloped virions inside vesicles, as well as partially enveloped virions and capsids in the cytoplasm. Often, there were several enveloped particles inside a single cytosolic vesicle (Figure 7B and Figure 8A, stars). These multiple particles in a single vesicle were very rarely seen in Becker-infected neurons (Figure 5A–C and Figure 6A,B). Again, we observed enveloped virions inside cytosolic vesicles that also contained electron-dense material surrounding or projecting from one pole of the capsid (Figure 7A and Figure 8A,B). This material might be tegument given that the particle near the center of Figure 8B clearly showed the electron-dense material extending from the tegument layer coating the capsid.

We counted cytoplasmic and cell surface virus particles. BaBe-infected hippocampal neurons showed less cell surface enveloped virions, in this case 11-fold fewer than Becker-infected neurons (Table 2). This number was larger than that involving CAD cells (7-fold), perhaps reflecting the point described above that CAD cells contain some undifferentiated cells. As well, BaBe-infected neurons showed marked accumulation of cytoplasmic enveloped particles (68% of the total), partially enveloped particles (7.7%) and unenveloped capsids (17%), so that most virus particles (93%) accumulated in the cytoplasm, rather than on cell surfaces. The two misassembled forms of enveloped particles described above, as well as increased numbers of partially enveloped and unenveloped particles were consistent with defects in the assembly of PRV BaBe particles. Moreover, there was a substantial accumulation of enveloped particles in the cytoplasm consistent with defects of sorting to cell surfaces and, likely but untested here, into axons. 

**Hippocampal neurons infected with a PRV mutant lacking US9, which expresses gE/gI, accumulated unenveloped capsids and enveloped virions in the cytoplasm.** We extended these studies to another PRV mutant that lacks only US9 and expresses gE, gI and US2. PRV 161 is a recombinant virus produced by creating a 258 bp deletion that removed most of the US9 coding sequences in PRV Becker, leaving the US8 (gE) and US7 (gI), US2 and other US genes (gD and gG) intact [3]. Importantly, PRV 161 produced normal plaques on MDBK epithelial cells, whereas BaBe produces substantially smaller plaques related to the loss of gE/gI [3]. As with BaBe and other US9-null mutants, PRV 161 failed to spread anterograde in the nervous systems of rats, but could spread normally in the eye and retrograde in neurons. Hippocampal neurons infected with PRV 161 displayed very few cell surface particles and more numerous cytoplasmic particles (Figure 9A–C). In Figure 9A, there were 2 or 3 cell surface particles and numerous cytoplasmic enveloped virions (empty arrow) and unenveloped capsids. There were also cytosolic vesicles that contained numerous virus particles and electron-dense material (Figure 9A, star). Figure 9B and the higher magnification Figure 10B show more numerous unenveloped (filled arrows) and partially enveloped capsids (asterisk in Figure 9B). Figure 9C and Figure 10B showed examples of enveloped virions in cytoplasmic vesicles with electron-dense material at one pole of the capsid. This electron-dense material appeared to be extensions of the virion tegument (Figure 10B). Counts of virus particles showed that PRV 161-infected neurons accumulated fewer cell surface particles (5.2%) and more cytoplasmic enveloped (74%), partially enveloped particles (5.5%), and unenveloped capsids (16%) (Table 2). There was a 14-fold decrease in cell surface particles with PRV 161 compared with wild-type Becker. Thus, it was clear that the PRV US9-null mutant 161, which lacks US9 and not gE/gI, was at least as defective in assembly and sorting as BaBe, a mutant lacking both US9 and gE/gI.

**Hippocampal neurons infected with PRV US9 mutants display lower quantities of infectious virus.** Our EM experiments would predict that PRV US9 mutants should produce less infectious virus within neurons, although many enveloped virions were observed in the cytoplasm. However, the imaging predicted more pronounced reductions in infectious virus particles outside neurons with the PRV US9 mutants. We infected rat hippocampal neurons with PRV Becker, BaBe and PRV 161 and harvested cell associated virus and, separately, virus present in cell culture supernatants at 12 h and 20 h. Infectious virus was tittered using porcine kidney PK15 cells. In preliminary experiments, we found that there was little infectious virus at 2 h after removing the input virus and washing the cells twice (not shown). Given that hippocampal neurons were expensive to purchase and consisted of only a million cells per vial, we were not able to characterize numerous time points of infection for three viruses and three replicates of each time point. After 12 h of infection, we found that there was a 5.9-fold reduction in cell associated infectivity comparing PRV 161 with wild-type Becker and a 5.2-fold reduction with BaBe at 12 h of infection (Table 3). As expected, the overall quantities of infectious virus present in the growth media (cell culture supernatants) of these neurons were approximately 10-fold lower than the cell-associated virus. Significantly, there were larger reductions in extracellular virus, comparing BaBe (8.2-fold) and PRV 161 (9.8-fold) with wild-type Becker (Table 3). We also observed a significant decline in infectivity between 12 and 24 h of infection in both the cells and media, perhaps related to cytotoxicity caused by PRV. We concluded that PRV US9-null mutants produced less infectious virus in the cytoplasm of neurons and more profound reductions in infectious virus in extracellular infectious virus. We note that Diwaker et al. observed less than 2-fold reductions in infectious virus in differentiated CAD neurons, comparing a different PRV gE/gI-US9-mutant with wild-type PRV. They used different methods and CAD neurons, not hippocampal neurons. As we noted above, not all CAD neurons differentiate into neuron-like cells. 

**Unenveloped PRV capsids in hippocampal axons/dendrites.** Hippocampal neurons produce numerous axons and dendrites (together denoted neurites) [30,31]. These neurites often made up a substantial fraction of our EM images because the neurites remained attached to neuron cell bodies along with the Thermanox plastic coverslips that was sectioned in these experiments. It has been reported that PRV is exclusively transported in axons as enveloped virions (Married particles) and not unenveloped capsids (Separate particles) in SCG neuron axons [4,28,29,34]. For HSV, there is now extensive evidence that, depending upon which neurons are studied, either enveloped virions or unenveloped capsids or both can be transported in axons (reviewed in [2]). We observed enveloped virions within the neurites of rat hippocampal neurons (not shown). However, we also observed examples of unenveloped capsids present in the axons/dendrites (Figure 11A,B filled arrows). This capsid can be compared to enveloped particles in the same section (empty arrows). The capsid shown was apparently associated with cellular membranes but does not appear to be present in a membrane vesicle. The attachment of capsids onto membranes might explain the mechanism by which unenveloped and enveloped virus particles can both be transported by kinesin motors (reviewed in [2]. In both cases, the motors might be attached onto membranes containing viral membrane and tegument proteins. There were insufficient numbers of PRV particles in neurites to allow counting of these particles. However, these observations suggest that, like HSV, PRV might also transport both unenveloped capsids and enveloped virus particles in neurites. One major caveat is that these neurons produce both axons and dendrites, so the capsids we observed might represent dendrites and not axons. 

## 4. Discussion

Anterograde axon transport includes at least three steps: (i) cytoplasmic assembly of virus particles, (ii) cytoplasmic sorting of particles into axons (which might involve kinesin motors), and (iii) kinesin-mediated transport on microtubule within axons. Our characterization of a HSV gE-/US9-double mutant supported models in which gE/gI and US9 function in the cytoplasm of neurons and are not required for transport in axons [8,21]. In the cytoplasm, there were major defects in assembly of HSV particles and the cytoplasmic sorting of particles to cell surfaces, as well as into axons. However, the few virus particles that are assembled and sorted into axons are transported normally. Similarly, a PRV mutant sorted very few virus particles into axons, but those that entered axons moved with normal kinetics and without major delays in transport [15]. Together, those studies argued that the transport of HSV and PRV particles in axons does not rely on either gE/gI or US9. Nevertheless, there have been reports, some that are recent, that HSV and PRV US9 binds to kinesins and this affects the velocity of anterograde transport within axons [16,17,19].

This debate about whether PRV US9 can function in axons to promote anterograde transport prompted us to investigate whether PRV US9 and gE/gI mutants exhibit defects in cytoplasmic assembly and sorting. Our observations were similar to those we reported for HSV [21]. In CAD and hippocampal neurons both PRV BaBe and PRV 161 assembled fewer enveloped particles and more unenveloped and partially enveloped capsids accumulated in the cytoplasm. Additionally, we observed examples of misassembled enveloped virions: multiple virions inside a single cytosolic vesicle or with electron-dense material protruding from one pole of the particle. However, there was also substantial accumulation of what appeared to be relatively normal enveloped virions in the cytoplasm. Associated with this cytoplasmic accumulation of virus particles there was a 7–15-fold decrease in cell surface particles in CAD and hippocampal neurons. Therefore, in addition to the defects in assembly, there were defects in sorting of virus particles to cell surfaces and into the media. It seems highly likely that these defects in sorting of virus particles to cell surfaces are closely related to defects in sorting of viruses into axons.

BaBe, which lacks gE/gI and US9, and PRV 161, which lacks just US9, showed similar defects in assembly and cell-surface sorting and, thus, we concluded that it is principally PRV US9 that promotes the neuron-specific assembly and sorting of virus particles. However, there have been numerous reports that HSV gE/gI participates in assembly and intracellular sorting in both neurons and epithelial cells [2,8,10,35]. We did not characterize a PRV mutant lacking just gE/gI and so we cannot comment on whether PRV gE/gI plays a more minor role in these assembly and sorting steps. However, it seems evident that the majority of the defects in PRV anterograde axonal transport result from loss of US9 producing defects in assembly and cytoplasmic sorting of virus particles.

Consistent with the EM data, we observed reductions in infectious virus in the cytoplasm of PRV US9-mutants. There were 5–6-fold reductions in infectious virus in the cytoplasm of neurons infected with PRV US9-mutants compared with wild-type PRV. There were also 8–10-fold reductions in infectious virus present in cell culture supernatants, suggesting more pronounced defects associated with reduced virus egress. Most of the previous characterization of these PRV US9 mutants in terms of infectious progeny involved non-neuronal cells. However, there were more limited observations of reduced yields of PRV US9-mutants following infection of rat sympathetic ganglia neurons. In one example, Tomashima and Enquist [6] observed similar quantities of a PRV null mutant at 16 h, compared with wild-type PRV. However, at 24 h there was 5-fold less infectious virus in cell-associated virus [6], similar to our results. We note that these authors concluded differently, that there were no defects in cell-associated infectivity. As we described above, we could not use SCG neurons in our EM experiments as these neurons produced too few virus particles for our imaging experiments. Thus, we could not directly compare our results with the previous results.

One might potentially argue that our conclusions require that we characterize repaired versions of PRV 161 and BaBe, i.e., viruses with restored expression of the gE/gI and US9 genes. The intent there would be to eliminate the possibility that there are other mutations apart from the mutations in the US9 and gE/gI genes in PRV161 and BaBe. However, we characterized two PRV mutants lacking US9 that were constructed independently and found very similar results. It would be extremely improbable that both viruses acquired second site mutations (outside the US9 and gE gene regions) that produced defects in virus assembly and sorting specifically in neurons. PRV 161 exhibited no defects in virus replication in PK15 cells and this virus could spread normally between epithelial cells, producing wild-type plaques on MDBK epithelial cells [3]. BaBe produces smaller plaques related to loss of gE/gI, but that is expected as gE/gI participates in epithelial spread. Moreover, both PRV 161 and BaBe could spread normally in the rat eye and in the retrograde direction in the central nervous system, but failed to spread anterograde [3]. Therefore, it is not plausible that both viruses acquired other mutations (distant from the US9 and gE genes) that produce defects in anterograde transport, as well as virus assembly and sorting specifically in neurons. Moreover, both these mutants were previously used to produce conclusions that US9 and gE/gI participate in anterograde transport within axons [4,18,23,35]. Our use of these mutants demonstrated that these same mutant viruses exhibited defects upstream processes, i.e., assembly and cytoplasmic sorting. 

We previously observed that loss of HSV gE and US9 leads to destabilization of the UL16 tegument protein [22]. It is possible that PRV UL16 was unstable in neurons infected with the PRV US9 mutants. Loss of UL16 might explain defects in assembly as UL16 is involved in envelopment. As well, reduced levels of UL16 might reduce sorting of enveloped virus particles into axons, though the connection between UL16 and sorting is not well established. If loss of UL16 is important in these defects, the effects of US9 might be, to some extent, indirect.

Our characterization of these PRV mutants demonstrated that US9 plays an important role in the assembly of enveloped virions in the neuronal cytoplasm and the sorting of these enveloped particles to cell surfaces. The reductions of virus particles within axons with PRV US9-null mutants are similar in magnitude to the reductions in cell surface particles (7–15-fold). Thus, it is likely that loss of US9 has global effects on both assembly and sorting of nascent PRV particles to cell surfaces and into axons and that these defects largely explain the observed defects in anterograde spread. Our experiments with HSV and with PRV do not exclude the possibility that US9 binds to kinesins in the cytoplasm in order to promote sorting of virus particles into axons as suggested [36]. Our results here, coupled with those of Daniel et al. who showed that loss of US9 did not reduce the velocity of PRV particles present in axons [15], argue quite strongly against models in which PRV US9 promotes kinesin-mediated transport within axons [16,17].

## Figures and Tables

**Figure 1 viruses-15-00153-f001:**
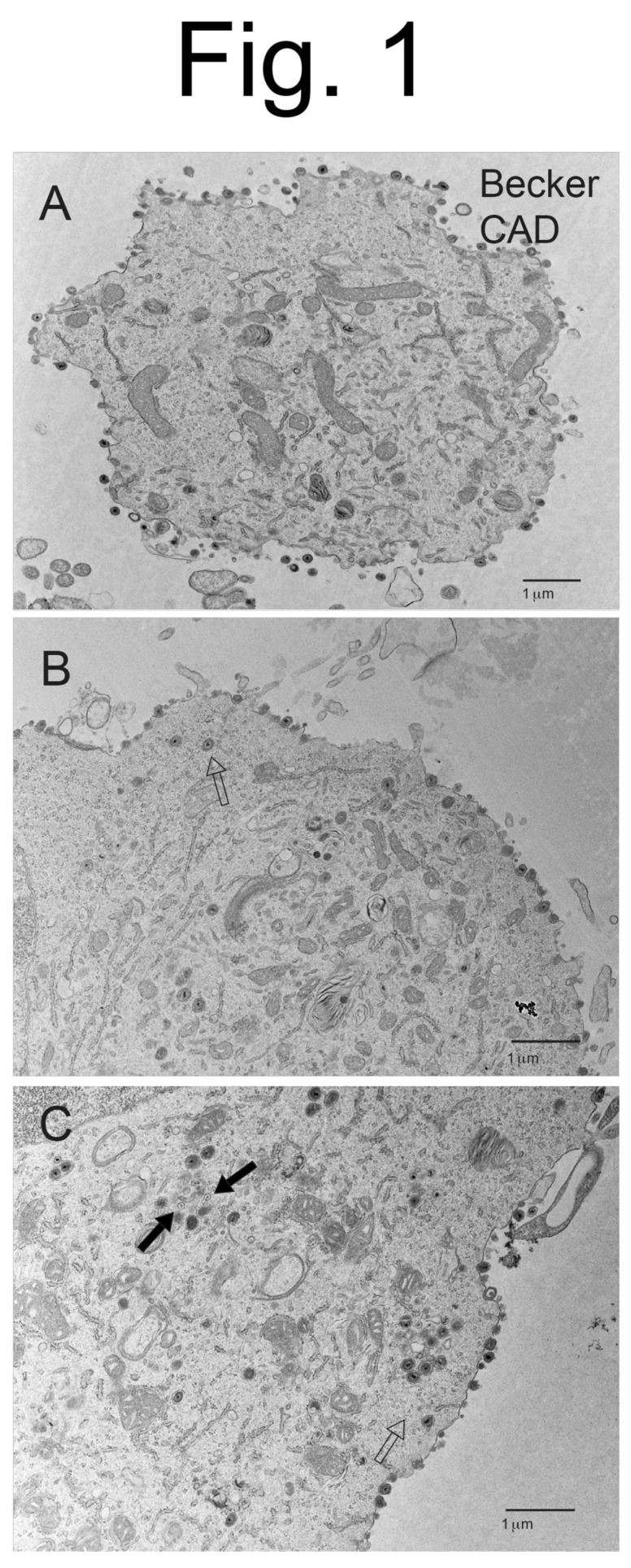
**Electron micrographs of CAD neurons infected with wild-type PRV Becker.** Differentiated CAD neurons growing on poly-lysine/laminin coated glass coverslips were infected with wild-type PRV Becker for 10–12 h. The cells were then fixed and processed for electron microscopy (EM). Panel (**A**) shows a section of the extremity of a neuron, without an obvious nucleus, that exhibited numerous cell surface particles and fewer cytoplasmic particles. Panel (**B**,**C**) shows numerous cell surface particles, as well as fewer cytoplasmic enveloped particles (empty arrow) and unenveloped capsids (solid arrows).

**Figure 2 viruses-15-00153-f002:**
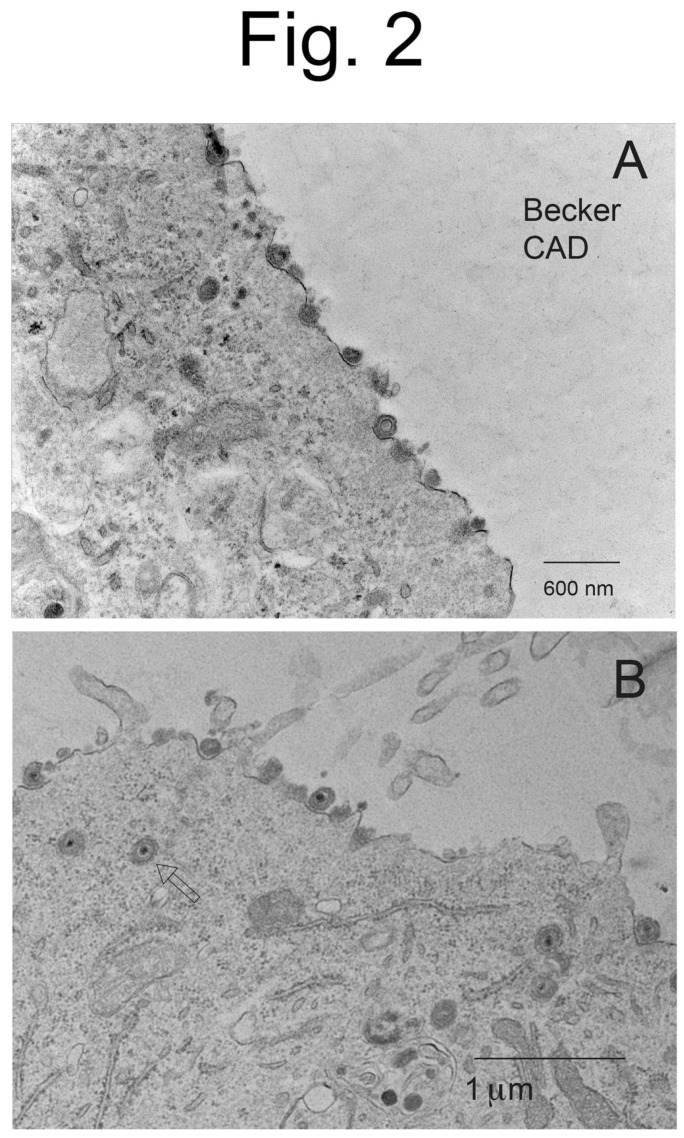
**Higher magnification EM images of CAD neurons infected with wild-type PRV Becker.** CAD neurons were infected with PRV Becker as in Figure 1 and processed for EM. Panel (**A**) shows cell surface virions. Panel (**B**) shows cell surface and cytoplasmic enveloped particles (empty arrow), which appeared of normal morphology.

**Figure 3 viruses-15-00153-f003:**
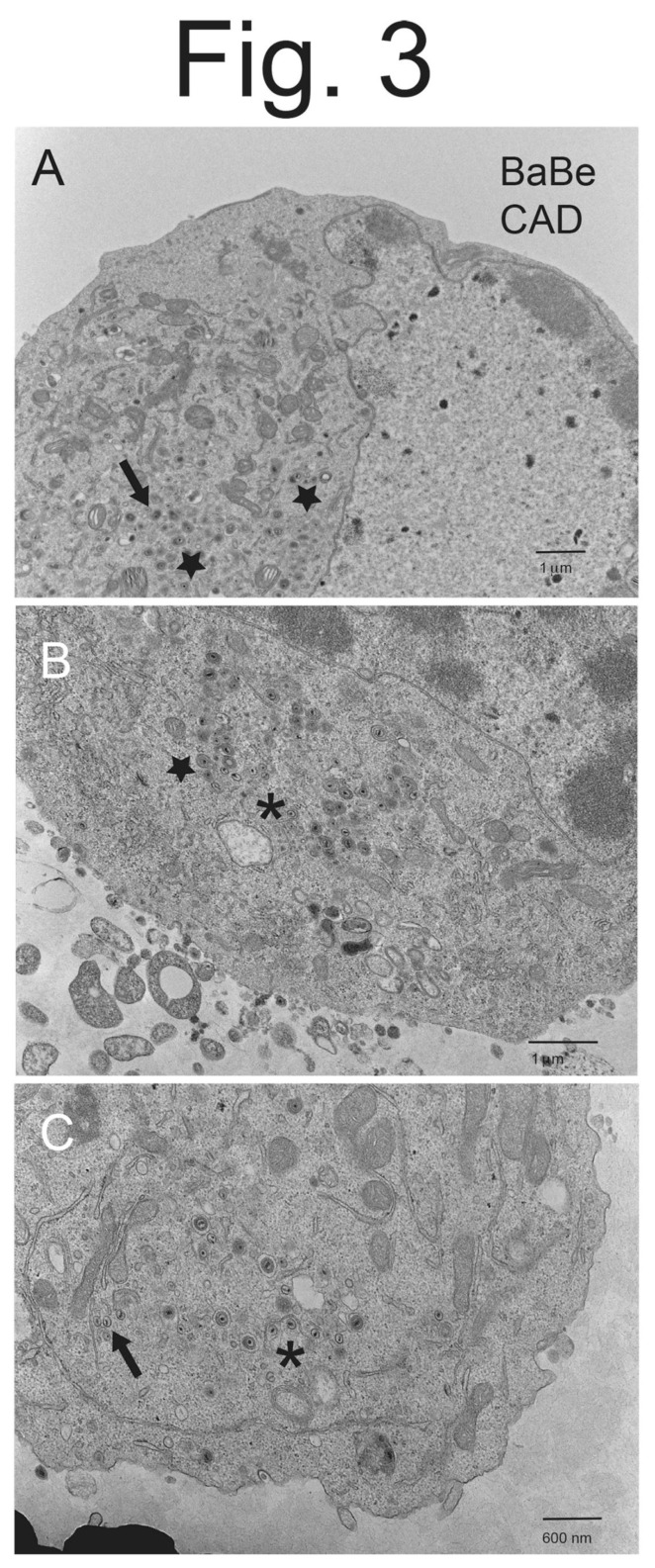
**Electron micrographs of CAD neurons infected with PRV BaBe.** Differentiated CAD neurons were infected for 10–12 h with PRV BaBe then fixed and processed for EM. Panel (**A**) shows a neuron with few cell surface particles and numerous enveloped cytoplasmic particles, including several that included electron-dense material inside the cytosolic vesicle that might be tegument (stars). There were also numerous unenveloped or partially enveloped capsids (solid arrow). Panel (**B**) shows some cell surface particles that appear misassembled as well as a partially enveloped capsid (asterisk) and misassembled virus particles with an oblong electron density (star). Panel (**C**) shows partially enveloped capsids (asterisk) and unenveloped capsids (solid arrow).

**Figure 4 viruses-15-00153-f004:**
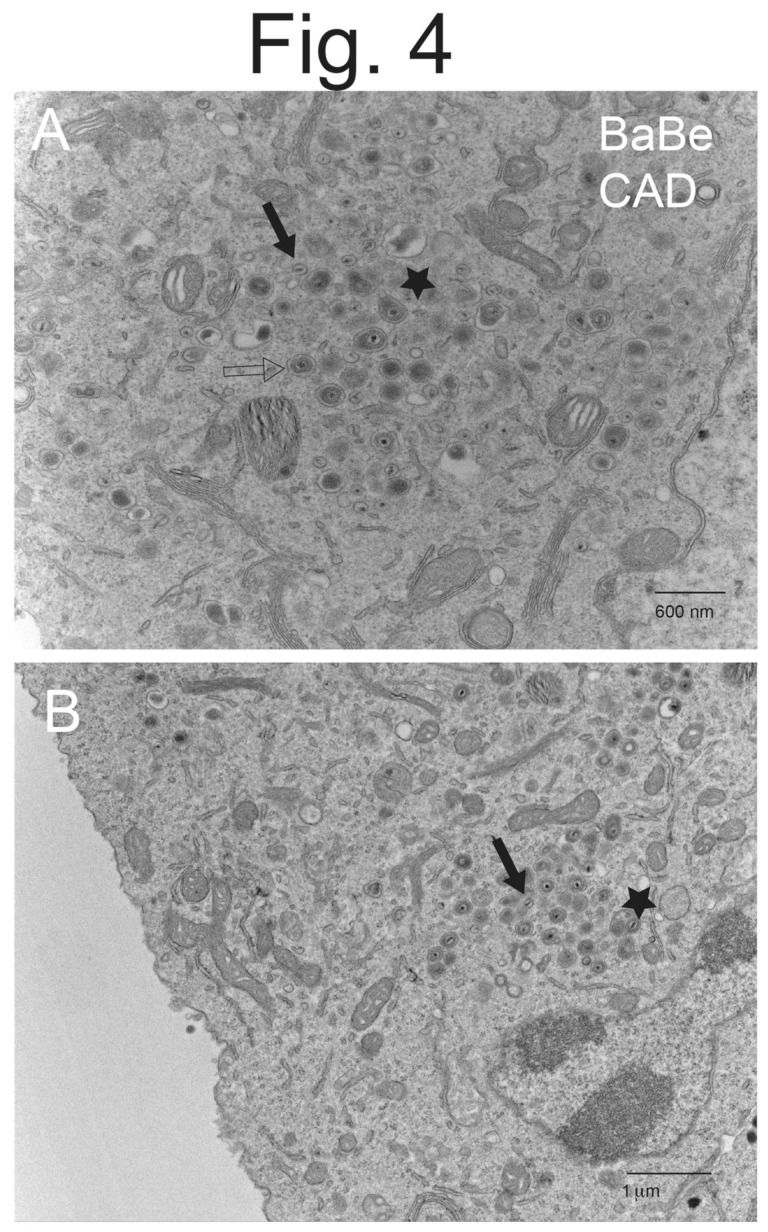
**Higher magnification EM of CAD neurons infected with PRV BaBe.** CAD neurons were infected with BaBe as in Figure 3 and processed for EM. Panel (**A**) shows fully enveloped virus particles in cytoplasmic vesicles (empty arrow), unenveloped capsids (solid arrow) and misassembled enveloped particles with oblong shapes and electron-dense material associated with the capsids (star). Panel (**B**) shows unenveloped capsids (solid arrow) and cytosolic vesicles with several virus particles (star).

**Figure 5 viruses-15-00153-f005:**
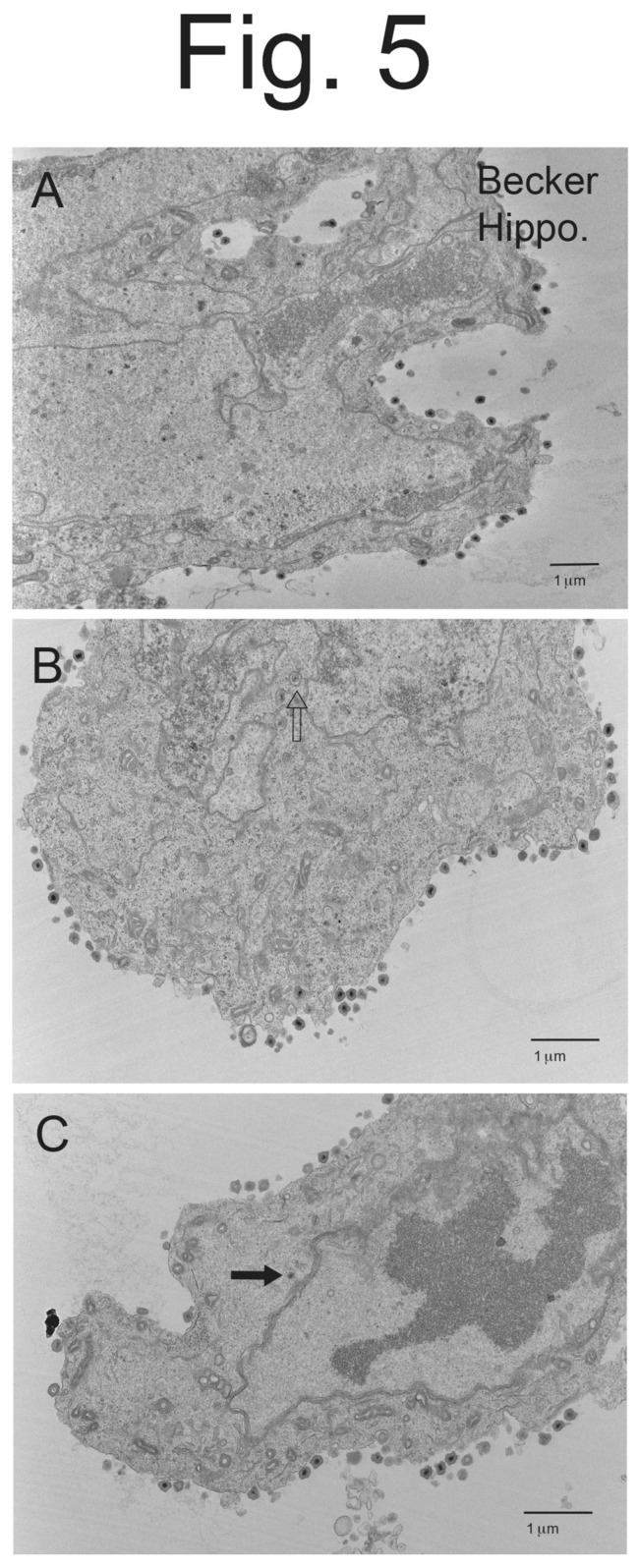
**Electron micrographs of rat hippocampal neurons infected with wild-type PRV Becker.** Hippocampal neurons were plated on lysine/laminin-coated Thermanox coverslips and infected with PRV Becker for 10–12 h then fixed before processing the Thermanox blocks with cells for EM. Panel (**A**) shows ample quantities of enveloped virions on the surfaces of neurons. Panel (**B**) also shows numerous surface enveloped virions, as well as rarer examples of partially enveloped and enveloped capsids (empty arrow). Panel (**C**) shows cell surface enveloped particles as well as an unenveloped capsid in the cytoplasm (solid arrow).

**Figure 6 viruses-15-00153-f006:**
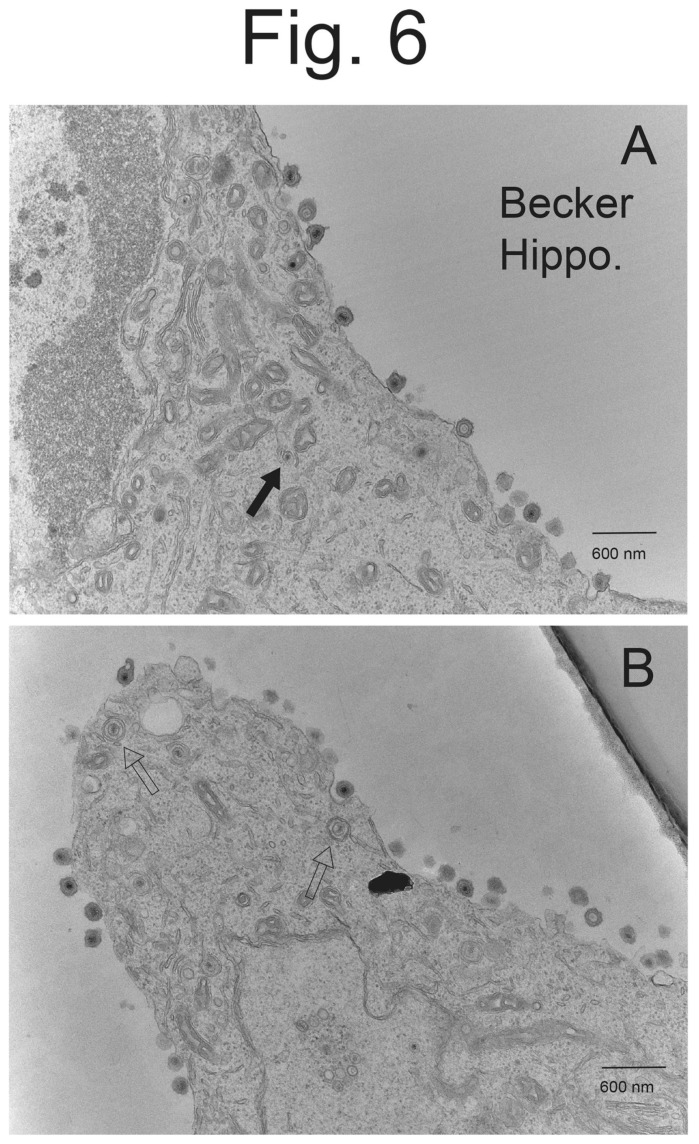
**Higher magnification EM of hippocampal neurons infected with PRV Becker.** Hippocampal neurons on Thermanox slides were infected with PRV Becker for 10–12 h then processed for EM. Panel (**A**) shows largely enveloped cell surface virions. There was one capsid in the cytoplasm associated with a membrane (solid arrow). Panel (**B**) shows cell surface virions and fewer cytoplasmic enveloped and partially enveloped virions (empty arrows).

**Figure 7 viruses-15-00153-f007:**
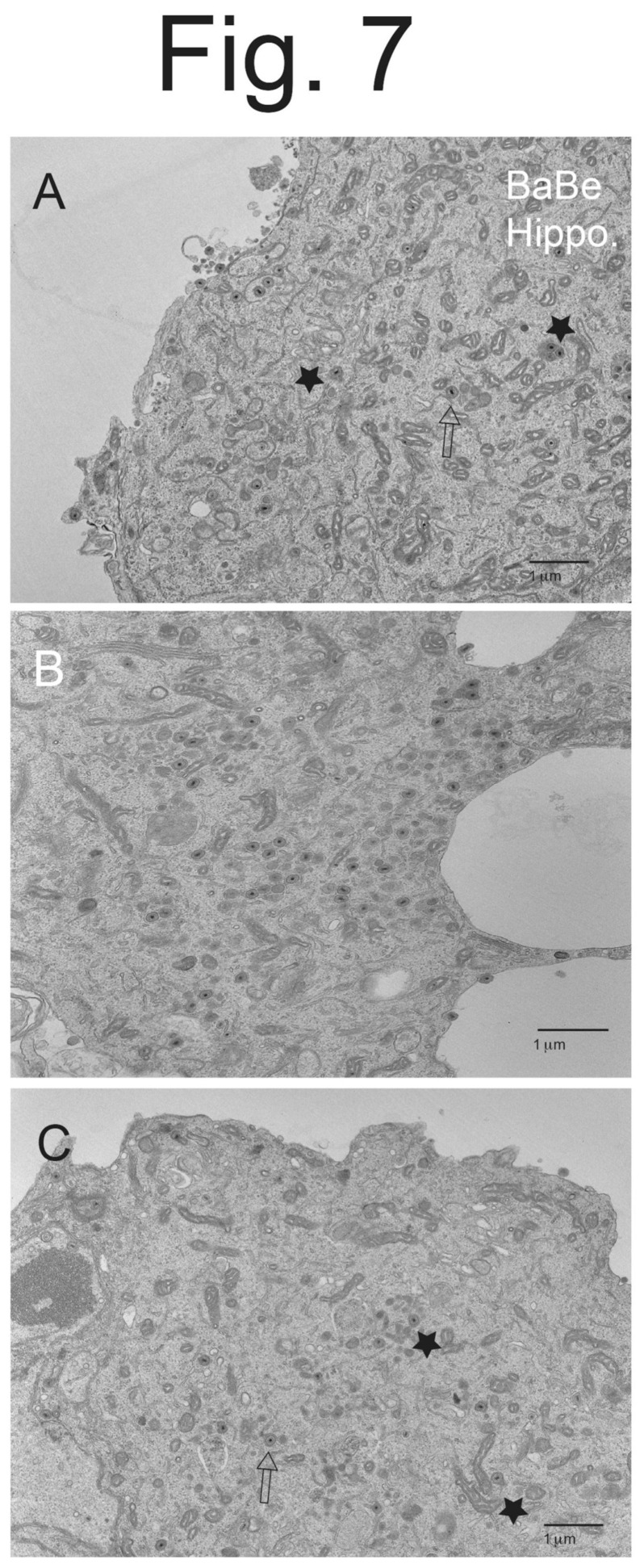
**EM images of hippocampal neurons infected with PRV BaBe.** Hippocampal neurons on Thermanox coverslips were infected with PRV BaBe for 10–12 h then processed for EM. Panel (**A**) shows a few enveloped particles on the surface of a neuron and more enveloped virions in the cytoplasm (empty arrow), as well as examples of misassembled virions (stars). Panel (**B**) shows numerous enveloped virions in the cytoplasm of a neuron, some that appear misassembled. Panel (**C**) shows few surface virions, cytoplasmic enveloped virions (empty arrow), and misassembled virions with electron-dense protrusions (stars).

**Figure 8 viruses-15-00153-f008:**
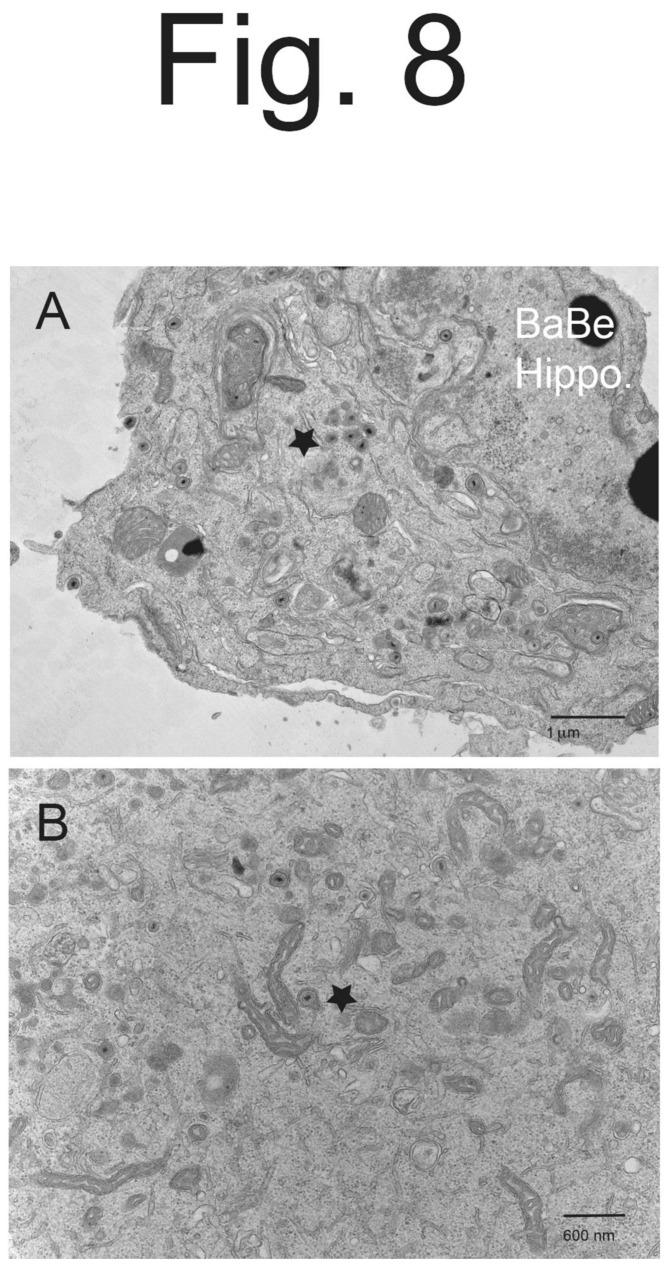
**Higher magnification EM of hippocampal neurons infected with PRV BaBe.** Hippocampal neurons on Thermanox coverslips were infected with PRV BaBe then processed for EM. Panel (**A**) shows numerous cytoplasmic virions that were misassembled, in some cases with numerous virions inside a single vesicle (star). Panel (**B**) shows a cytosolic enveloped virion with electron-dense material extending from one pole of the capsid (star). There were also several partially enveloped capsids in this panel.

**Figure 9 viruses-15-00153-f009:**
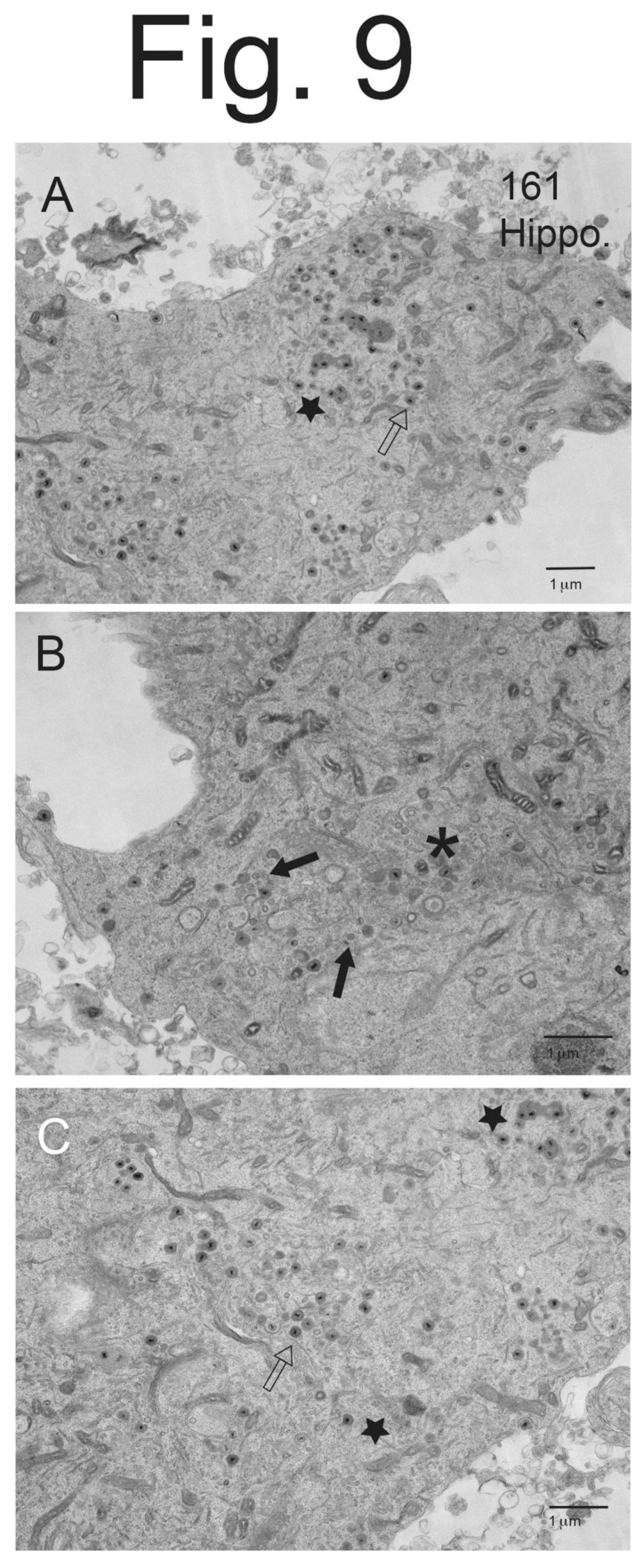
**EM images of hippocampal neurons infected with PRV 161.** Hippocampal neurons on Thermanox slides were infected with PRV 161 for 10–12 h then processed for EM. Panel (**A**) shows a neuron with few cell surface particles and large numbers of cytoplasmic enveloped particles (empty arrow), as well as enveloped particles that were misassembled with more than a single capsid in a vesicle and often immersed in electron-dense material. * Panel (**B**) shows a region of cytoplasm with numerous unenveloped capsids (filled arrows). There were also several partially enveloped capsids (asterisk). Panel (**C**) shows a neuron with numerous cytoplasmic enveloped virions that appeared normal (empty arrow) and others that showed evidence of electron-dense material at one pole or several capsids in a single vesicle (stars). * Star in Panel (**A**).

**Figure 10 viruses-15-00153-f010:**
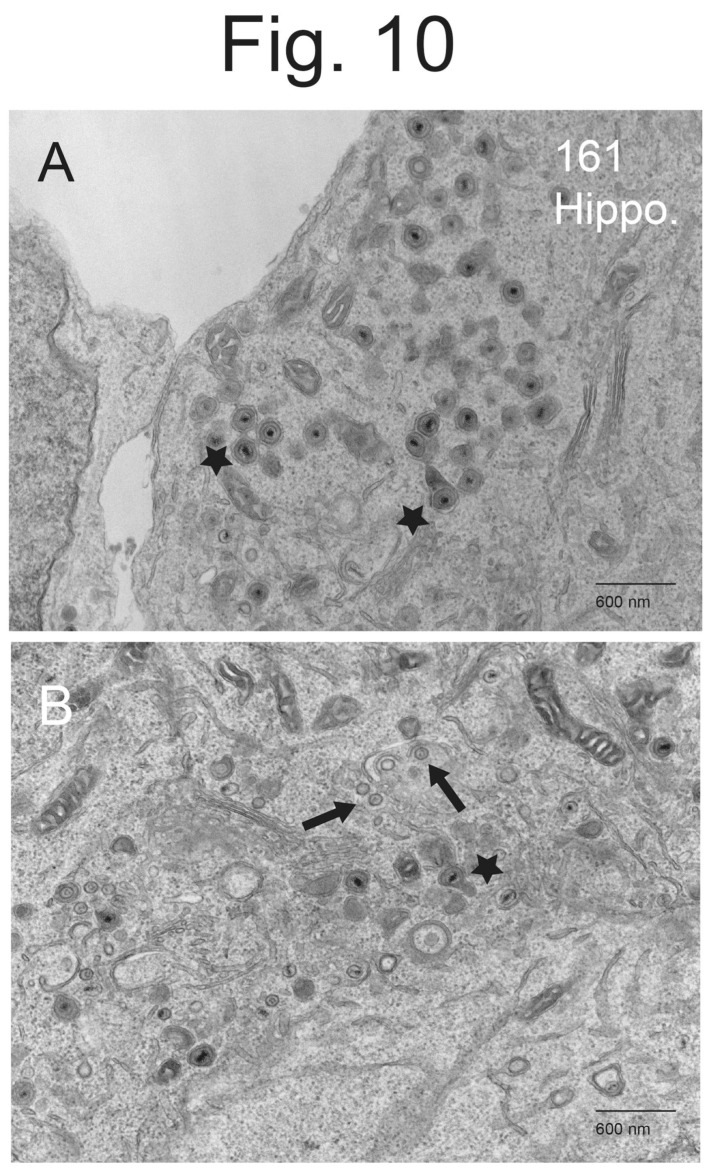
**Higher Magnification EM of Hippocampal Neurons Infected with PRV 161.** Hippocampal neurons on Thermanox slides were infected with PRV 161 then processed for EM. Panel (**A**) shows enveloped virus particles in the cytoplasm and no particles on cell surfaces. There were clear examples of vesicles that contained more than a single virus particle and a cytosolic enveloped virus with an oblong appendage of electron-dense material extending from one pole of an enveloped virion (stars). Panel (**B**) represents a higher magnification of panel Figure 9B showing numerous cytoplasmic enveloped capsids, some that were bound to membranes and partially enveloped (solid arrows). There was also an example of an enveloped virus particle with an oblong protrusion of electron-dense material (star) and numerous unenveloped capsids.

**Figure 11 viruses-15-00153-f011:**
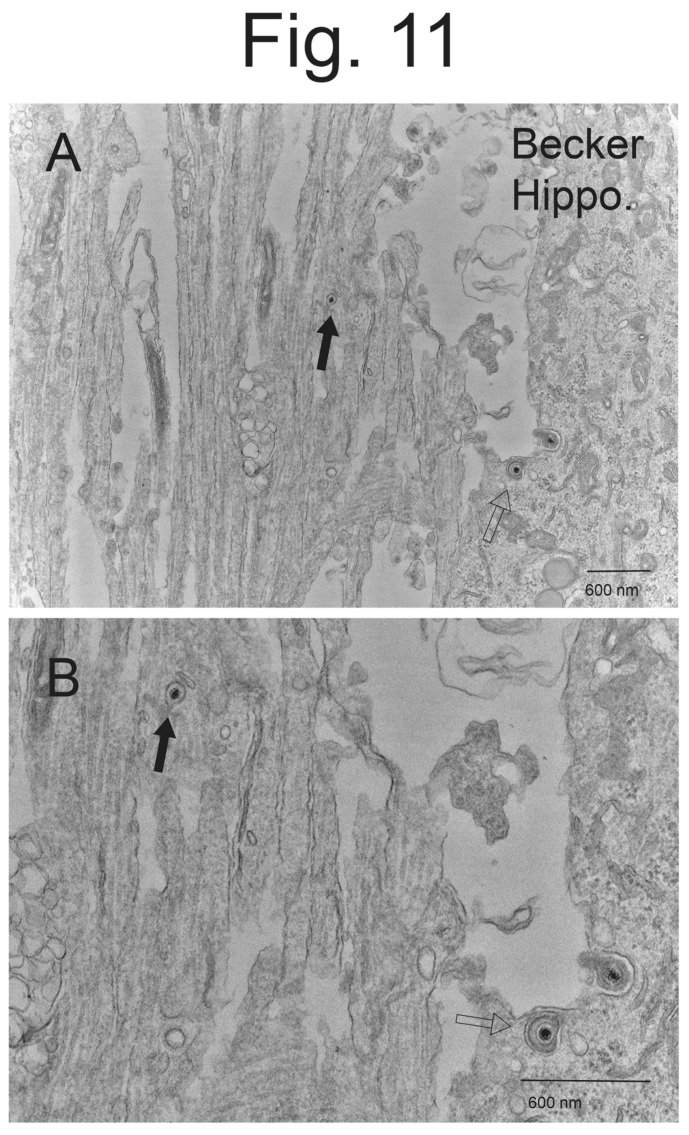
**EM images of a hippocampal neurite containing a PRV unenveloped capsid.** Panel (**A**) shows a PRV unenveloped capsid inside a rat hippocampal neuron neurite (solid arrow). This particle can be compared to enveloped virus particles in the cytoplasm of a nearby neuron (open arrow). Panel (**B**) shows a higher magnification of these particles. It appears that the unenveloped capsid is attached onto a membrane vesicle inside the neurite (solid arrow).

**Table 1 viruses-15-00153-t001:** **Cytoplasmic and cell surface forms of PRV particles associated with CAD neurons**.

	Surface Enveloped	Cyt. Enveloped	Partially Enveloped	Unenv. Capsids
*Becker*	880 (77%)	192 (17%)	29 (2.5%)	46 (4.0%)
*BaBe*	155 (11%)	902 (65%)	103 (7.4%)	224 (16%)

**Table 2 viruses-15-00153-t002:** **Cytoplasmic and cell surface forms of PRV particles associated with hippocampal neurons**.

	Surface Enveloped	Cyt. Enveloped	Partially Enveloped	Unenv. Capsids
*Becker*	835 (75%)	197 (18%)	38 (3.4%)	50 (4.5%)
*BaBe*	101 (7.3%)	942 (68%)	106 (7.7%)	234 (17%)
*PRV 161*	66 (5.2%)	941 (74%)	70 (5.5%)	203 (16%)

**Table 3 viruses-15-00153-t003:** **Infectious virus produced by hippocampal neurons infected with PRV mutants**.

	Cell-Associated Infectious Virus		Extracellular Infectious Virus	
	12 h	20 h	12 h	20 h
*Becker*	3.3 × 10^4^ ± 0.9	1.3 × 10^4^ ± 1.2	4.1 × 10^3^ ± 0.8	2.2 × 10^3^ ± 0.5
*BaBe*	6.3 × 10^3^ ± 0.7	2.0 × 10^3^ ± 1.0	5.0 × 10^2^ ± 0.6	2.1 × 10^2^ ± 0.6
*PRV 161*	5.6 × 10^3^ ± 0.6	1.6 × 10^3^ ± 0.9	4.2 × 10^2^ ± 0.7	1.6 × 10^2^ ± 0.3

## Data Availability

Not applicable.

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
