# Peer review of "Pseudorabies Virus Mutants Lacking US9 Are Defective in Cytoplasmic Assembly and Sorting of Virus Particles into Axons and Not in Axonal Transport"

_viruses, 2023, doi:10.3390/v15010153_

Round 1

Reviewer 1 Report

Using TEM, Adamou, Vanarsdall and Johnson have investigated the assembly of PrV strains lacking gE/gI and US9, or US9 alone, in the cytoplasm of cultured rodent neurons. Similar to what has been observed for HSV gE/gI/US9 mutants, the authors conclude that PrV US9 plays a critical role in the assembly and transport of enveloped virions to the cell surface. Specifically, accumulations of non-enveloped capsids as well as enveloped virions are observed in the cytoplasm of neurons infected with mutant strains that are accompanied by reductions in the appearance of extracellular enveloped virions on the cell surface. Additionally, aberrantly assembled virions, such as multiple capsids within a single envelope, were observed in neurons infected with mutant strains. The functions of gE/gI and US9 in anterograde spread of alphaherpesviruses have been a subject of intense investigation for over two decades and the findings produced from multiple laboratories have not always been congruent. The data described in this manuscript provide significant new information that will be of substantial interest to herpes virologists and may inform new models of US9 function. The manuscript is well written, the data are of high quality and, for the most part, the data shown are in alignment with the conclusions.

Major concern:

In their Introduction, the authors reference their study indicating that cells infected with HSV mutants lacking gE/gI and Us9 have reduced levels of the tegument protein UL16. Interestingly, the phenotype of HSV UL16 null strains is very similar to the phenotypes of the US9 mutants described in this manuscript (e.g. accumulations of non-enveloped capsids in the cytoplasm as well as formation of enveloped virions containing multiple capsids). Could the defects in virion assembly observed by the authors be due to reduced levels UL16 in these cells? In other words, is US9 promoting UL16 stability rather than functioning directly in virion envelopment as the authors have suggested? This alternative interpretation was not discussed in the manuscript. Additionally, the authors should consider examining UL16 levels in cells infected with Becker, BaBe, or PRV 161.

Reviewer 2 Report

Herpes simplex virus (HSV) is involved in various mucocutaneous and neurological diseases. Although anti-viral drugs are available for herpes, these drugs cannot completely eradicate the virus from the host as HSV can establish late infection in neurons. How HSV can enter into CNS and come back from it after reactivation is one of the most important question for HSV. Thus, retrograde and anterograde transport of HSV virions in axon has been studied by many groups. From the analysis using knock out viruses, HSV gE/gI and Us9 are considered to play important roles during anterograde transport. However, the authors’ group recently reported that these proteins function during assembly or sorting in the cytoplasm but not in the axon. In this manuscript, the authors analyzed the gE/gI and US9 triple deletion PRV and the Us9 deletion PRV to reveal both mutants exhibited similar defects in virus assembly and cytoplasmic sorting of virus particles to cell surfaces. In agreement with the previous observations, the phenotypes of the triple mutant and Us9 single mutant were similar in neurons. Thus, the authors concluded that PRV Us9 primarily functions in neurons to promote cytoplasmic steps before axonal transport during anterograde transport.

This manuscript is well written and will be of interest to the broader readership of this journal. However, I have a few comments listed below to improve this manuscript before publication.

(i) In this manuscript, the authors analyzed using electron microscopy. However, using this method, we cannot analyze the velocity of virions. As described in the past, interaction between Us9 and kinesins are considered to effect on velocity of virions in axons (lines 356-357). Thus, in the final conclusion, the authors should note that the effect of Us9 on velocity of virions was not analyzed in this manuscript.

(ii) Even though the results in this manuscript suggested that Us9 is dispensable for axonal transport, the authors should discuss how HSV or PRV virions are efficiently transported in axons. We considered that gE or Us9 is important in this process for many years and this is why there are many papers focusing on gE or Us9.

(iii) In table 3, the titers of PRV were reduced during 12h-20h and low. It seems that PRV cannot replicate in these cells. As these authors analyzed anterograde transport or assembly of the viruses, the cells must support viral replication. The authors should add the viral titer at other time point or in the presence of polymerase inhibitors.

(iv) Fig 4 and 8 should be enlarged to reveal budding defect at star easily for the readers.
